# The Bilateral Interplay between Cancer Immunotherapies and Neutrophils’ Phenotypes and Sub-Populations

**DOI:** 10.3390/cells11050783

**Published:** 2022-02-23

**Authors:** Naomi Kaisar-Iluz, Ludovica Arpinati, Merav E. Shaul, Sojod Mahroum, Mohamad Qaisi, Einat Tidhar, Zvi G. Fridlender

**Affiliations:** 1Institute of Pulmonary Medicine, Hadassah Medical Center, Jerusalem 91120, Israel; naomi.kaisar@mail.huji.ac.il (N.K.-I.); ludovica.arpinati@mail.huji.ac.il (L.A.); meravsha@hadassah.org.il (M.E.S.); sojod.m97@gmail.com (S.M.); mohamad.qaisi@mail.huji.ac.il (M.Q.); einatid@gmail.com (E.T.); 2Faculty of Medicine, Hebrew University of Jerusalem, Jerusalem 91120, Israel

**Keywords:** neutrophils, lung cancer, immunotherapy, PD-L1, PD-1, G-CSF

## Abstract

Immunotherapy has become a leading modality for the treatment of cancer, but despite its increasing success, a substantial number of patients do not benefit from it. Cancer-related neutrophils have become, in recent years, a subject of growing interest. Distinct sub-populations of neutrophils have been identified at advanced stages of cancer. In this study, we aimed to evaluate the role of neutrophils in mediating the efficacy of immune checkpoint inhibitors (ICI) treatments (α-PD-1/PD-L1), by assessing lung tumor models in mice. We found that G-CSF overexpression by the tumor significantly potentiates the efficacy of ICI, whereas neutrophils’ depletion abrogated their responses. Adoptive transfer of circulating normal-density neutrophils (NDN) resulted in significantly reduced tumor growth, whereas low-density neutrophils (LDN) had no effect. We next investigated the effect of ICI on neutrophils’ functions. Following α-PD-L1 treatment, NDN displayed increased ROS production and increased cytotoxicity toward tumor cells but decreased degranulation. Together, our results suggest that neutrophils are important mediators of the ICI treatments and that mainly NDN are modulated following α-PD-L1 treatment. This research provides a better understanding of the function of neutrophils following immunotherapies and their impact on the efficacy of immunotherapy, supporting better understanding and future improvement of currently available treatments.

## 1. Introduction

Immunotherapy and immune modulation have become central to leading efforts in the research and practice of cancer treatment. However, despite the increasing success of immunotherapy, a substantial number of patients do not benefit from it [1,2]. Tumor-associated neutrophils (TAN) have become, in recent years, a subject of growing interest in cancer research [3,4]. The main role of neutrophils resides in the innate immune response and clearance of infection as the first line of defense [5,6,7]. However, neutrophils also represent a significant portion of the immune infiltrate in the tumor microenvironment. It has been shown that the presence of neutrophils in the circulation and in the tumor is associated with poor prognosis in many types of cancer [8,9] and affects various aspects of cancer biology [3,10]. The importance of neutrophils in human cancer has been clearly demonstrated as neutrophilia and high neutrophil-to-lymphocyte ratio (NLR) were found to be associated with poorer prognosis in many cancers [11,12]. Furthermore, NLR has been connected to lower response to immunotherapy treatments such as α-PD-1/PD-L1 and α-CTL-4 [13,14,15]. In recent years, sub-populations of neutrophils have been discovered. Normal-density neutrophils (NDN) can be found in the circulation of both healthy donors and cancer patients, while low-density neutrophils (LDN) can be found mostly in chronic inflammation and cancer [16,17]. The amount and presence of LDN in mice and humans significantly increases with tumor growth. LDN actually consist of two distinct populations: mature and immature neutrophils [17,18]. NDN are more prone to present anti-tumor properties, whereas LDN are more prone to present a pro-tumor phenotype. NDN display cytotoxic capacities and an ability to increase tumor cell death, while LDN appear to promote a supportive microenvironment for tumor development, displaying reduced cytotoxicity towards tumor cells [16,19]. In addition, LDN can have suppressive capacities towards other cells of the immune system, hence relating to them frequently as part of the myeloid-derived suppressor cells (MDSCs), specifically the Polymorphonuclear (PMN)-MDSCs [20]. In the tumor microenvironment (TME), tumor-associated neutrophils (TAN) can demonstrate anti-tumor (N1) or pro-tumor (N2) phenotype. TAN’ polarization towards an N2 or N1 phenotype is driven by cytokines such as TGF-β and IFN-β, respectively [21,22,23]. N2 TAN accumulate during tumor progression and contribute to tumor growth [24].

Immune checkpoint inhibitors (ICI) have shown to be very effective treatments in various types of cancers and blockade of the PD-1/PD-L1 axis can result in dramatic and sustained tumor regression [25,26]. PD-1 is expressed on T cells, and negatively regulates their antitumor effect, while and PD-L1 is expressed on various human cancers such as lung cancer, breast cancer, and melanoma, as well as on tumor-infiltrating immune cells in the tumor microenvironment [25]. On neutrophils, PD-L1 expression is up-regulated in patients with hepatocellular carcinoma and negatively correlated with patients’ survival [27]. PD-L1+ neutrophils were shown to suppress T-cell function and contribute to human gastric cancer progression in vivo [28]. It is possible that modulating neutrophils’ phenotype and function could help improve the efficacy of current immunotherapies. For example, it has been demonstrated in animal models of lung and breast cancer that the addition of a TGF-β blocking agent increases the efficacy of α-PD-1/PD-L1 treatments [29,30].

Another potential player in the bilateral interplay between neutrophils and ICI is G-CSF. G-CSF is used routinely for recovery from neutropenia in patients undergoing chemotherapy, since it promotes the release of neutrophils from the bone marrow [31]. Ina addition to its impact on neutrophils’ recruitment to the circulation, G-CSF has been suggested to promote neutrophils’ phagocytic and anti-bacterial activity [32] and enhance their ROS production [33]. G-CSF has been shown to induce both pro- and anti-tumor effects, by affecting both tumor cells and immune cells [4]. In this work, we aimed to evaluate the bilateral effects of α-PD-L1/PD-1 immunotherapies and neutrophils` phenotypes and sub-populations in murine lung tumor models, with emphasis on the role of G-CSF and the differential effects of ICI on the different neutrophils’ subpopulations.

## 2. Materials and Methods

### 2.1. Animals

Six to eight weeks old, 20–25 gr C57Bl/6 mice were purchased from Harlan Laboratories (Jerusalem, Israel). For the LKRM model, 129/SVJ mice, purchased from Jackson Laboratories were bred with C57Bl/6 mice and the first generation only was used in experiments. Mice were housed under specific pathogen-free conditions at the Hebrew University School of Medicine Animal Resource Center. All experiments were approved by the Animal Research Committee of the Hebrew University School of Medicine. 

### 2.2. Cell Lines and Tumor Injection

The following tumor cell lines, LLC (Lewis Lung Carcinoma), LLC-luciferase, LKRM (Lung K-Ras G12D mutant Modified), were cultured and maintained in DMEM media, supplemented with 10% heat-inactivated fetal bovine serum, 2 mM glutamine, 100 U/mL penicillin, and 100 μm/mL streptomycin (all from Biological Industries, Kibbutz Beit-Haemek, Israel). Cell cultures were maintained at 37 °C and 5% CO_2_. LLC and LLC-luciferase cell lines were purchased from ATCC. LKRM cell line was a kind gift from Prof. Steven Albelda, University of Pennsylvania, Philadelphia, PA, USA. G-CSF has been described as an important factor driving the release of neutrophils from bone marrow [31], but also capable of affecting myeloid cells’ phenotype [34,35] and modulating the tumor microenvironment [36,37]. In order to assess the impact of G-CSF on tumor growth and immune checkpoint blockade efficacy, we compared the original LLC cell line and an LLC-GCSF-transfected cell line (referred to as LLC-GCSF which secretes high amounts of G-CSF). LLC-GCSF-transfected cell line was obtained following transfection with murine G-CSF, which was cloned into GFP-expressing MIGR1 lentiviral vector using XhoI and EcoRI restriction sites to generate viruses. Twenty hours post infection, LLC-infected cells were subsequently selected and isolated by sorting the GFP-positive population using a flow a BD FACS Aria III sorter. The cells were cultured as a G-CSF enriched cell line. All cell lines were regularly tested and maintained negative for mycoplasma contamination; 1–2 × 10^6^ LLC or LLC-GCSF tumor cells were injected subcutaneously to the flank of C57Bl/6 mice; 1 × 10^6^ LKRM tumor cells were injected subcutaneously to the flank of B6/129 mice tumor growth was measured every 1–3 days, and tumor volume was calculated using the formula volume = length × width^2^ × 3.14/6. In order to assess the change in growth rate and to normalize between multiple experiments, tumor growth ratio was calculated based on tumor size on each day vs. size on day 0 per each mouse.

### 2.3. Murine Neutrophils Isolation from Blood and Tumor

Tumor-bearing mice were sacrificed when subcutaneous primary tumors reached a volume of 200–800 mm^3^. Blood was collected by cardiac puncture using insulin syringes previously rinsed with heparin. Blood was diluted in sterile PBS containing 0.5% BSA and gently loaded onto a discontinuous Histopaque gradient (1.119 and 1.077, Sigma, St. Louis, MO, USA) and centrifuged at 700× *g* for 30 min, RT, no brake. The Low-density fraction (LDF) was collected from the plasma–1.077 interface (containing LDN, monocytes and lymphocytes) and normal-density neutrophils (NDN) were collected from the 1.077–1.119 interface. Neutrophil purity was determined by FACS and was consistently found to be >95% for NDNs. RBC lysis was performed with water for 30 s and stopped by adding 5 × PBS supplemented with 2.5% BSA. LDNs were isolated from the LDF using EasySep Mouse Neutrophil Enrichment kit (STEMCELL Technologies). Following isolation, LDNs purity was determined via FACS and was consistently found to be >95%. Flank tumors were harvested, minced and digested at 37 °C for 1 h in L15 medium (Sigma) supplemented with 0.05 mg/mL collagenase type II mg/mL, 0.025 mg/mL Elastase (both from Worthington Biochemical, Lakewood, NJ, USA) and 0.025 mg/mL DNase I (Roche applied science, Rotkreuz, Switzerland) at 37 °C for 45 min. The red blood cells were lysed using RBC lysis buffer (Biological Industries). Tumor-associated neutrophils (TAN) were isolated from tumors by staining the whole tumor cell suspension with PE-conjugated α-Ly6G antibody (Biogems, Westlake Village, CA, USA) and then applying the EasySep PE positive Selection Kit (STEMCELL Technologies, Petach-Tiqva, Israel) according to the manufacturer’s protocol. Purity was tested by FACS and was consistently found to be >85%.

### 2.4. Splenic Cells Isolation

Spleens from LLC tumor-bearing mice were harvested, minced into HBSS buffer (Biological Industries) supplemented with 2% FBS and centrifuged at 300× *g* for 10 min at RT. RBCs were lysed using ACK buffer (0.15 M NH4Cl, 10 mM KHCO3 and 0.1 mM Na2 EDTA) and neutralized by adding 10 mL of RPMI-1640 medium. The cells were filtered through a 40 μm filter and centrifuged at 300× *g* for 10 min at RT. The pellet was resuspended in RPMI supplemented with 10% heat-inactivated fetal bovine serum, 100 U/mL penicillin and 100 μg/mL streptomycin.

### 2.5. Treatments and Injections

Treatments were started when LKRM or LLC tumors reached a size of ~200 mm^3^. Adoptive transfer of neutrophils was carried out on LKRM tumor-bearing mice, injected with 2–5 × 10^6^ NDN or 5 × 10^6^ LDN isolated from LLC-GCSF tumor-bearing mice. The mice were injected intraperitoneally (IP) for 3 days in a row. LKRM and LLC tumor-bearing mice were injected IP with 250 μg α-PD-L1 or α-PD-1 antibodies (BioXCell (Lebanon, NH, USA), clone RMPI-14, clone 10F.9G2, respectively) every 3 days for a total of 3 injections. LKRM and LLC tumor-bearing mice were treated IP with 150µg α-Ly6G antibody (BioXCell, clone 1A8) for depletion of neutrophils every 2 days for a total of 4 injections. Depletion was confirmed using flow cytometry. Control mice were injected IP with either PBS or Isotype Rat IgG2A (BioXCell, clone 2A3) or IgG2B (BioXCell, clone LTF-2). LLC tumor-bearing mice were injected IP with MPO inhibitor 4-ABAH (4- Aminobenzoic hydrazide 95%, Sigma Aldrich) at a dose of 40 μg/g in 500 μL of HBSS (Biological Industries), twice a day, starting from day 8 following tumor injection for 12 days.

### 2.6. Staining and Flow Cytometry

Following isolation cells were resuspended in FACS buffer and incubated with “FcBlock” (anti-mouse CD16/CD32 BioLegend, San Diego, CA, USA) for 15 min on 4 °C. Cells were then stained for 45 min on ice for various markers. Antibodies used include a-Ly6G-PE/APC/BGViolet450/FITC (Biogems), a-Ly6G-Violet blue (Miltenyi Biotec, Gaithersburg, MD, USA), a-CD8-BGViolet450 (Biogems), and a-PD-L1-PE (Biogems). Immunostained cells were analyzed with a BD LSR-Fortessa cell analyzer using FlowJo X software.

### 2.7. In Vitro Killing Assay

LLC Luciferase tumor cells (1 × 10^4^ cells/well) were plated in 100 μL RPMI-1640 (Sigma) with 10% FBS placed in each well of a white 96- flat-bottom well plate (Greiner bio-one). Then, 1 × 10^5^ of murine NDN/LDN/TAN were added to the tumor cell culture to reach a ratio of 1:10 (tumor cells: neutrophils) in the absence or presence of 10 µg/mL of α-PD-L1 or α-PD-1 antibodies and co-cultured overnight. Following incubation, tumor cell death was assessed via measurement of luciferase activity, using the Luciferase Assay System (Promega, Madison, WI, USA) according to the manufacturer’s instructions. Luciferase activity was measured using a Tecan F200 microplate luminescence reader. The percentages of the killing were calculated by the formula [1-(co-culture LLC-luc + neutrophils/only LLC-luc)] × 100.

### 2.8. ROS Production

Purified NDN/LDN/TAN were resuspended at a concentration of 1 × 10^3^/µL in Hank’s balanced salt solution without phenol red (HBSS) and placed in each well of a white 96-flat-bottom well plate (Greiner bio-one). Following incubation in the absence or presence of 10 µg/mL α-PD-L1/α-PD-1 for 4 h at 37 °C, a luminol + Horseradish peroxidase (HRP) (both from Sigma-Aldrich) stock solution in HBSS was added to each well to a final concentration of 50 μM luminol and 40 g/mL HRP. Cells were then treated with 50 nM PMA (Sigma) and chemiluminescence was determined for 1 h at 30 s intervals, using a Tecan plate reader (InfiniteF200Pro, TECAN, Männedorf, Switzerland ).

### 2.9. In Vitro T-Cell Proliferation

Splenocytes were obtained from LLC tumor-bearing mice as described above. Cells were diluted in PBS (10 × 10^6^ cells/mL) and incubated with 2.5 µM CFSE (Molecular Probes, Invitrogen) for 15 min at 37 °C in the dark. Labeling was quenched by adding an excess of warm RPMI medium supplemented with 10% FBS and incubated for 5 min at 37 °C in the dark. The cells were then centrifuged at 300× *g* for 10 min at RT. 0.25 × 10^6^ cells were plated in RPMI/proliferation media (10% heat-inactivated FBS, 10 mM HEPES, 50 µM β-ME, 200 µM L-glutamine) in 96-well plates and stimulated with 1 µg/mL anti-CD3 (Biogems, clone 145-2C11) and anti-CD28 (BioLegend, clone 37.51) antibodies. NDN/LDN/TAN were added to CFSE labeled splenocytes at a ratio of 1:1, in the absence or presence of 10 µg/mL α-PD-L1/α-PD-1. The level of CD8 T cell proliferation was determined after 72 h by FACS using anti-CD8α antibody (Biolegend). 

### 2.10. Degranulation of Elastase 2

Next, 0.5 × 10^6^ of NDN/LDN/TAN were incubated in the absence or presence of 10 µg/mL α-PD-L1/α-PD-1 in RPMI for 3 h at 37 °C. Then, 50 nM PMA was added to the cells for another hour at 37 °C. The cells were centrifuged at 300× *g* for 10 min at 4 °C and the supernatants were collected. The levels of elastase 2 (ELA2) in the supernatants were determined by enzyme-linked immunosorbent assays (ELISA) according to the manufacturer’s instructions (R&D Systems, Inc., Shanghai, China).

### 2.11. Statistical Analysis

Statistical analyses were performed using GraphPad Prism software version 8. Comparisons between two groups were carried out using a one/two-tailed unpaired/paired *t*-test as appropriate. For comparisons of more than two groups we used ANOVA multiple comparison with proper post hoc corrections. Differences were considered significant when *p* < 0.05.

## 3. Results

### 3.1. G-CSF Overexpression by the Tumor Potentiates the Efficacy of ICI

We aimed to assess the impact of G-CSF on tumor growth in the LLC model and appraise whether the presence of G-CSF could modify the efficacy of immune checkpoint blockade in this model. Since we did not observe any difference in the rate of tumor growth between the two models (Figure 1A), we further evaluated the impact of PD-L1 and PD-1 blockade on tumor growth in both lines in vivo (Figure 1B–E). α-PD-1 treatment did not impact LLC tumor growth, and α-PD-L1 treatment only moderately impaired tumor growth in this model (*p* < 0.05, Figure 1B) compared to the untreated mice. In contrast, both α-PD-1 and α-PD-L1 treatments strongly delayed LLC-GCSF tumor growth in vivo (*p* < 0.0001, Figure 1C). The robust response of LLC-GCSF tumor-bearing mice resulted in a greater decrease in tumor volume following one week of treatment with both α-PD-L1 (*p* < 0.01, Figure 1D) and α-PD-1 (*p* < 0.001, Figure 1E), compared to LLC tumor-bearing mice. Interestingly, the effect of the treatments could be observed already at day 6 in the LLC-GCSF model (*p* ≤ 0.0002), unlike in the LLC model in which only a minor delay in tumor growth was noted in the treated mice on day 7.

As a response to α-PD-L1 therapy, we found a reduction in the preponderance of LDN in the circulation of LLC-GCSF tumor-bearing mice and a reduction in the expression of PD-L1 on TAN (Appendix A). We further evaluated the impact of PD-L1 and PD-1 blockade in vivo on tumor growth in the LKRM model (a murine tumor model presenting low infiltration of neutrophils). Although the treatments drove a significant decrease in LKRM tumor growth ratio (Appendix A), this effect was not as robust as the one seen in the LLC-GCSF model and was not reflected by a decrease in tumor size following one week of treatment (Appendix A). Overall, these results suggest that G-CSF may potentiate the efficacy of α-PD-1 ad α-PD-L1 immunotherapies.

### 3.2. Neutrophils Participate in the Response to ICI

In order to evaluate whether the neutrophils play a role in mediating the effect of α-PD-L1 immunotherapy, we depleted neutrophils in LLC-GCSF tumor-bearing mice, using an in vivo α-Ly6G antibody in combination with α-PD-L1. As previously shown by us and others [38,39], neutrophils’ depletion resulted in a significant reduction in tumor growth to a similar extent as PD-L1 blockade alone (based on growth ratio, *p* < 0.001 and tumor size *p* < 0.01). However, neutrophils’ depletion in combination with α-PD-L1 abolished this reduction in tumor growth and eliminated the response to the checkpoint blockade (Figure 2A). At day 7, a significant decrease in tumor weight following each treatment separately was recorded (*p* < 0.05, Figure 2B) in contrast to the combined treatment (α-PD-L1 + α-Ly6G), which resulted in a similar tumor size to the non-treated group. 

Depletion of neutrophils was confirmed based on Ly6G levels inside the tumors by using FACS (Figure 2C). Since these results suggest that neutrophils participate in the effect of ICI, we evaluated the impact of the combination of α-PD-L1/α-PD-1 with α-MPO on tumor growth in both LLC models. Myeloperoxidase (MPO) is an inflammatory enzyme that is most abundantly expressed in azurophilic granules of neutrophils [40]. MPO blockade in combination with ICI in LLC-GCSF diminished the response to these treatments, and no significant difference could be seen in tumor growth compared to the untreated group (Appendix A). However, MPO blockade in combination with α-PD-1 in LLC potentiated the efficacy of the immunotherapy (Appendix A). These results support our findings that G-CSF at least partially potentiate the efficacy of ICI via neutrophils. Interestingly, blocking MPO did not abolish the response to the same extent as depletion did, suggesting that the effect of neutrophils is mediated not only by MPO, but by other processes as well.

### 3.3. G-CSF Modulates the Preponderance of Neutrophils in the Circulation and Tumors and Increases Neutrophils’ Expression of PD-L1

We next evaluated the impact of G-CSF on neutrophils’ phenotype, comparing different neutrophil subpopulations in LLC and LLC-GCSF tumor-bearing mice. G-CSF overexpression in LLC-GCSF mice resulted in prominent higher levels of LDN (35.59 ± 3.82% vs. 6.24 ± 1.28% *p* < 0.0001, Figure 3A) and TAN (32.36 ± 3.15% vs. 7.27 ± 1.07% *p* < 0.0001, Figure 3B) compared to LLC mice.

In addition, G-CSF drove a sizable increase in PD-L1 expression levels in LDN (73.83 ± 3.86 in LLC-GCSF vs. 12.37 ± 2.59 in LLC *p* < 0.0001 Figure 3C) and TAN (69.57 ± 2.77 in LLC-GCSF vs. 15.75 ± 2.02% in LLC *p* < 0.0001 Figure 3D). The expression level of PD-L1 in NDN was similarly high in both models (93.26 ± 2.03% vs. 95.45 ± 1.07% Figure 3E). Additionally, we assessed the proportion of TAN and LDN as well as PD-L1 expression levels in the LKRM model. LKRM mice presented lower percentage of LDN (Appendix A) and lower infiltration of TAN (Appendix A) compared to the LLC models. The expression of PD-L1 in LDN (Appendix A) and TAN (Appendix A) was lower in the LKRM compared to LLC-GCSF, but higher than LLC. These results suggest that G-CSF potentiate the response to ICI by driving the recruitment of neutrophils and increasing the expression of PD-L1 on neutrophils (LDN and TAN).

### 3.4. Adoptive Transfer of NDN Reduces Tumor Growth

We next aimed to evaluate the immunological consequences of manipulating the amount of circulating neutrophil subsets (NDN and LDN) on tumor development. For this purpose, we used the LKRM model, which is a murine cancer model naturally low in neutrophils both in circulation and in the tumor and applied adoptive transfer of either NDN or LDN. The injection of NDN (isolated from LLC-GCSF tumor-bearing mice) resulted in a significant reduction in LKRM tumor growth by means of growth ratio over three consecutive days (*p* < 0.001, Figure 4A Left panel), as assessed at day 3 (*p* < 0.05, Figure 4A Right panel) and based on growth rate per day (*p* < 0.05, Figure 4B). 

In contrast, injection of LDN did not demonstrate such an impact (Figure 4A,B). Following adoptive transfer of either NDN or LDN, a tendency toward increased neutrophil infiltration to the tumor (TAN) was observed, compared to the control mice (*p* = 0.06, Figure 4C). These results suggest a different impact of the different neutrophil subpopulations on tumor growth, with NDN exhibiting an anti-tumor effect, whereas LDN do not demonstrate such an effect.

### 3.5. PD-L1 Blockade Modifies NDN Functions

We next tested the effect of α-PD-L1/α-PD-1 treatments on neutrophils’ functions in vitro. First, we evaluated degranulation in the different sub-populations following exposure to ICIs, reflected by the level of elastase secreted to the media. NDN showed a significant decrease in degranulation following exposure to α-PD-L1 (*p* < 0.01, Figure 5A top panel), whereas LDN and TAN did not show any change in this parameter (Figure 5A middle and bottom panels, respectively). Following α-PD-L1 treatment, NDN were able to more efficiently kill cancer cells compared to control (*p* < 0.01, Figure 5B top panel), while LDN and TAN did not show any difference (Figure 5B middle and bottom panels). To further understand what could mediate this enhanced cytotoxicity toward tumor cells, we tested ROS production. NDN showed a significant increase in ROS production following incubation with α-PD-L1, compared to control neutrophils (*p* < 0.001, Figure 5C left top panel). Again, we could not observe any modulation in LDN’s and TAN’ ROS production following α-PD-L1 treatment (Figure 5C left middle and bottom panels, respectively). Interestingly, we did not observe any impact on these functions in any of the neutrophil subpopulations following α-PD-1 treatment (Appendix A). We next tested the impact of α-PD-L1 on neutrophils’ ability to suppress CD8+ T cells proliferation reflecting their function as suppressor cells. Whereas about 90% of T cells did not show any proliferation following isolation (Figure 6A left panel), CD8+ T cells activation by αCD3 and αCD28 strongly triggered proliferation, with 72.3% of the cells showing a high proliferation rate and 10.34% a moderate proliferation rate (Figure 6A right panel). Co-culture of the CD8+ T cells with NDN did not affect their CD8+ T cells proliferation (Figure 6B left panel). In contrast, co-culture with LDN suppressed T cell proliferation with a significant decrease in highly proliferated population (47.98% vs. 72.3%) and increase in intermediate proliferated (37.19% vs. 10.34%) (Figure 6B middle panel), as previously shown [16,41]. Interestingly, incubation with TAN mildly promoted CD8+ T cells proliferation according to an increase in the highly proliferated population (86.51% vs. 72.3%) and a decrease in lowly proliferated (5.787% vs. 17.34) (Figure 6B right panel).

Nevertheless, the addition of α-PD-L1 to the co-culture with any of the three neutrophil subpopulations did not change the modulations seen in CD8+ T cell proliferation, and there was no difference compared to the incubation with each of the neutrophil subpopulations only (Figure 6C). In conclusion, our data suggest that out of the three neutrophil subpopulations tested here, PD-L1 blockade impacts significantly and only the function of NDN, promoting an anti-tumor cytotoxic phenotype in this subset.

## 4. Discussion

In the current study, we evaluated the bilateral impact of blocking the PD-1/PD-L1 axis on neutrophils and the effect of neutrophils on these treatments. We investigated the impact of ICI in lung tumor models with a detailed comparison between different sub-populations of neutrophils. We first compared a G-CSF-overexpressing LLC model, which recruits more neutrophils to the circulation and the TME, to the original LLC cell line as well as the LKRM model, both of which are naturally low in neutrophils. Tumor-bearing mice were treated with PD-1/PD-L1 blockade, and LLC-GCSF tumor-bearing mice displayed much stronger responses than LLC and LKRM to these ICI (Figure 1 and Appendix A). Interestingly, mice bearing LLC-GCSF tumors had not only higher levels of neutrophils, but their neutrophils also expressed higher levels of PD-L1, compared to the LLC and LKRM models (Figure 3 and Appendix A). Importantly, neutrophils were found to be crucial in the effect of α-PD-L1, as their depletion abrogated the responses to it (Figure 2). Adoptive transfer of NDN (isolated from LLC-GCSF tumor-bearing mice) clearly showed that NDN possess anti-cancer ability in vivo, whereas LDN had no clinical effect on tumor growth, demonstrating the differential potential of these two subpopulations (Figure 4). Assessing the impact of α-PD-L1/α-PD-1 on neutrophils’ functions in vitro, NDN but not LDN showed better ability to produce ROS and kill cancer cells, though they secreted lower amounts of elastase (Figure 5). 

In our study we demonstrate that G-CSF is an important player in the bilateral interplay between neutrophils and immunotherapy. G-CSF has been described as an important factor that promotes the mobilization of neutrophils to the circulation [31], and also enhances neutrophils’ ROS production [33]. We have previously shown that incubation of healthy human neutrophils with G-CSF results in neutrophil extra-cellular traps (NETs) formation [42]. In this study, we show that tumor-derived G-CSF increases the amount of LDN and TAN and drives higher expression of PD-L1 in these neutrophils (Figure 3 and Appendix A). He et al. showed that neutrophils isolated from hepatocellular carcinomas displayed a significant increase in the expression of PD-L1 following exposure to GM-CSF and TNF-α [27]. Similar to our findings, a recent study by Rui Sun et al. demonstrated that tumor-derived G-CSF induces PD-L1 expression on neutrophils of colon cancer CT-26 tumor-bearing mice [43]. In our study we advance another step, showing that the efficacy of ICI is greater in lung cancer models secreting G-CSF vs. those that do not (Figure 1 and Appendix A), suggesting that this improvement is mediated via neutrophils.

An interesting finding we present is the difference between the positive effects of PD-L1 blockade vs. the lack of effect of PD-1 blockade in these models (Figure 1). Several studies have reported such a lack of response to α-PD-1 immunotherapy in LLC and K-RAS models [38,44,45,46]. We noticed, however, that LKRM tumor-bearing mice responded to PD-1 blockade somewhat better than LLC (Appendix A). In our study, the expression of PD-L1 on neutrophils (LDN and TAN) from the LKRM model was higher than in the LLC neutrophils (Appendix A). We therefore first hypothesized that the different effect could result from a positive correlation between the expression levels of PD-L1 on neutrophils and the efficacy of ICI treatments, i.e., if neutrophils express a higher level of PD-L1, the efficacy of ICI is enhanced. We could not find, however, clear changes induced by blocking PD-L1 in vitro in the functions of LDN and TAN we tested. Since we have recently shown that NDN can change in vivo to both LDN and TAN [47], it is possible that the changes found in the NDN functions are eventually seen in the other neutrophils’ subpopulations in vivo, affecting the results of ICI. In that context, it is important to remember that high levels of PD-L1 in the tumor is a well-known biomarker for the efficacy of ICI [48,49]. Interestingly, a recent case report described a patient with stage IV NSCLC producing high levels of G-CSF with a favorable response to Pembrolizumab (α-PD-L1) [50]. The authors suggested that Pembrolizumab monotherapy may be an effective treatment for patients with advanced G-CSF-producing NSCLC, a suggestion that our data presented herein support. 

High neutrophil-to-lymphocyte ratio (NLR) is known to be associated with poorer prognosis in many cancers [11,12]. In addition, high NLR has been connected to lower response to ICI, whereas a decrease in NLR values during treatment has been associated with better response [13,15,51]. Additionally, up-regulation of PD-L1 expression on neutrophils has been negatively correlated with patients’ survival [27]. We and others have shown that cytokines such as TGF-β can drive a phenotypical modulation of TAN from N1 (anti-tumor) to N2 (pro-tumor) [21,22,23], as well as modulating NDN to become LDN [16,17,47]. In parallel, it has been demonstrated that the addition of an anti-TGF-β agent increases the efficacy of ICI treatments in animal models of lung and breast cancer [29,30], although they did not assess the specific changes in neutrophils following this treatment. In the current study, we found that the effect of α-PD-L1 on tumor growth in the LLC-GCSF model was completely abrogated when the mice were depleted of neutrophils (Figure 2). This result demonstrated the importance of neutrophils in the response to α-PD-L1 treatment. Moreover, this finding enhances our hypothesis that G-CSF potentiates the efficacy of α-PD-L1 via its effect on neutrophils. In an opposite study, Rui Sun et al. reported that neutrophils’ depletion combined with α-PD-L1 improved survival of tumor-bearing mice and reduced tumor growth [43]. We believe that these conflicting results could be due to the different cancer models used, i.e., lung vs. colon cancer. Interestingly, there are additional reports showing that neutrophils’ depletion mildly but significantly enhanced the effect of α-PD-1 in an additive way, suggesting that neutrophils are not a major part of the mechanism of this therapy [38,52]. Additional evidence for the importance of neutrophils in mediating the efficacy of α-PD-1/PD-L1 therapies was brought herein by combining an MPO inhibitor with these treatments (Appendix A). In a similar manner to our results with neutrophils’ depletion in the LLC-GCSF model, we found that inhibition of MPO impeded the response to immunotherapies. In contrast, in the LLC model, inhibition of MPO with α-PD-1 resulted only in a tendency towards a reduction of tumor growth, as previously shown by Ugolini et al. [45]. 

As mentioned before, cancer-related neutrophils are known to be able to display both anti- and pro-tumor effects. For example, NDN present cytotoxic abilities to tumor cells, whereas LDN present immunosuppressive properties [16,19]. To assess the role of the different circulating neutrophils, we evaluated the differential impact of adoptive transfer of NDN and LDN on tumor growth in vivo. Injection of isolated NDN to recipient LKRM tumor-bearing mice resulted in decreased tumor growth and a significant response, whereas injection of LDN had no effect on tumor growth (Figure 4). Both NDN and LDN adoptive transfers induced a tendency to increase the percentage of TAN in the tumor (Figure 4), supporting our recent findings demonstrating the ability of both NDN and LDN to infiltrate tumors in vivo [47]. 

Based on the interesting findings of differences between the sub-populations of neutrophils, we evaluated in vitro the effects of PD-1/PD-L1 blockade on specific neutrophils’ functions. Following exposure to α-PD-L1, the ability of NDN to kill cancer cells was significantly improved compared to untreated cells (Figure 5). This is in agreement with a recent study showing the importance of the PD-L1/PD-1 axis in NDN cytotoxicity towards breast cancer cell lines [53]. Neutrophils secrete many factors to the microenvironment that can affect their pro- and anti-tumor effects [54,55]. We found that α-PD-L1-treated NDN secreted less elastase but more ROS than control untreated neutrophils (Figure 5). On one hand, ROS production by neutrophils at low levels can induce T-cell activation, proliferation and function. On the other hand, high levels of ROS can inhibit T-cell proliferation and ROS can induce DNA damage affecting cancer cell proliferation and epithelial damage, therefore supporting tumor progression [54,55,56]. It is possible that the increase in ROS secretion by NDN following PD-L1 blockade was part of the mechanism increasing NDN cytotoxicity towards tumor cells. Elastase is generally accepted to be more pro-tumor [57], although there is recent evidence that human (but not murine) neutrophils release catalytically active neutrophil elastase (ELANE) capable of killing many cancer cell types with minimal toxicity to non-cancer cells [58]. Elastase could play a pro-metastatic role via EMT and extracellular matrix (ECM) remodeling. It was found to degrade the anti-cancer protein thrombospondin 1 in the pre-metastatic tumor microenvironment to promote cancer progression [55]. By impairing neutrophil degranulation, PD-L1 blockade consequently diminishes the presence of proteases such as elastase, with possible pro-tumor properties in the tumor microenvironment. Our findings, therefore, suggest an additional mechanism through which neutrophils might mediate the effect of α-PD-L1 therapy.

As we have previously shown [16] and show again here, LDN but not NDN significantly reduced CD8+ T-cells proliferation (Figure 6). However, we did not find an impact of ICI in vitro on neutrophils’ modulation of CD8+ T cells proliferation (Figure 6). It is possible that the significant effects of α-PD-L1 on the functions of NDN with only mild to no effect on LDN and TAN is related to the higher expression (and lower variability) of PD-L1 by NDN, compared to LDN and TAN (Figure 3). 

In summary, in the current manuscript, we evaluate the bilateral effects of PD-L1/PD-1 blockade and neutrophils’ sub-populations. We found differences between the response to α-PD1 vs. α-PD-L1 of different lung cancer models. We found that tumor-derived G-CSF potentiate the efficacy of ICI, probably due to a subsequential increase in the expression of PD-L1 on the neutrophils. Additionally, we demonstrate that neutrophils are important for the response to PD-L1 treatment and may be involved in the mechanism of this treatment, possibly by affecting the functions of NDN. Our findings add another layer to the understanding of the complex mechanisms of ICI in cancer. One potential implication that must be further investigated is using G-CSF as an adjuvant therapy to ICI, possibly allowing the improved effect of α-PD-L1 therapy. As a first step, it is suggested to evaluate whether patients that receive G-CSF for other reasons have indeed an improved response to therapy. In addition, when using ICI, one has to keep in mind that myeloid cells in general and neutrophils specifically can be affected by ICI therapy, and more important can have an important influence on the efficacy of this therapy and, hence, the prognosis of the patients treated. Although our data focused on lung cancer models, it is possible that these findings can be expanded to other types of cancer that are treated with ICI. Further research must be carried out to evaluate such effects. In general, our findings support the importance of combining other strategies with PD-L1/PD-1 blockade in order to achieve better anti-cancer responses, possibly treatments targeting neutrophils and other myeloid cells.

## Figures and Tables

**Figure 1 cells-11-00783-f001:**
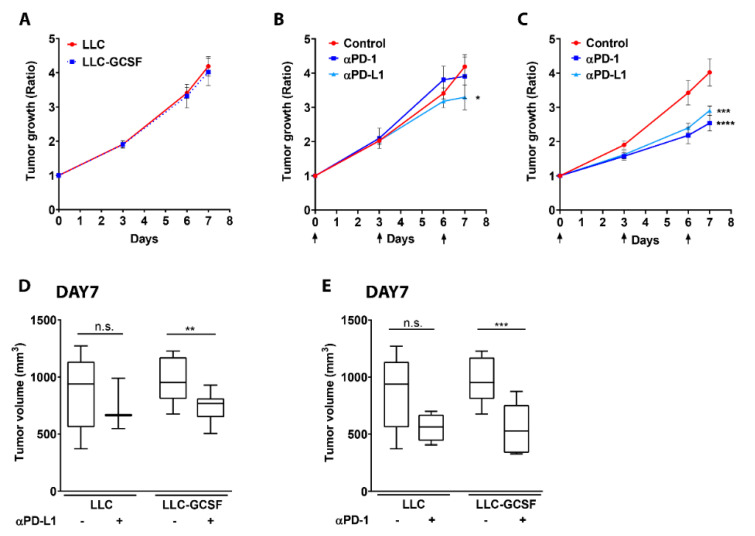
Impact of α-PD-L1 and α-PD-1 blockade on tumor progression of LLC vs. LLC-GCSF models. (**A**) LLC or LLC-GCSF cells were injected to the flank of mice. When tumors’ size reached 200–250 mm^3^, treatment was started (defined as day 0), and tumor size was monitored. Tumor growth ratio was calculated based on tumor size on each day vs. day 0. (**B**,**C**) Mice injected to the flank with either LLC (**B**) or LLC-GCSF (**C**) were treated with 250 µg α-PD-L1 or α-PD-1 antibodies. Days of treatments are marked by arrows. Tumor growth was monitored following treatments. (**D**,**E**) Tumor volume on day 7 for LLC and LLC-GCSF models is presented with or without α-PD-L1 (**D**) or α-PD-1 (**E**) treatment. Means values ± SE are shown, *n =* 3–12 in each group per experiment. Statistical differences between each treatment and the untreated control group are expressed with stars (*). Statistical significance was determined by one-way ANOVA with sidak’s post hoc test (**B**,**C**) or with unpaired two-tailed *t*-test (**D**,**E**), both with *p* < 0.05. * *p* < 0.05, ** *p* < 0.01, *** *p* < 0.001, **** *p* < 0.0001 and n.s—non-significant.

**Figure 2 cells-11-00783-f002:**
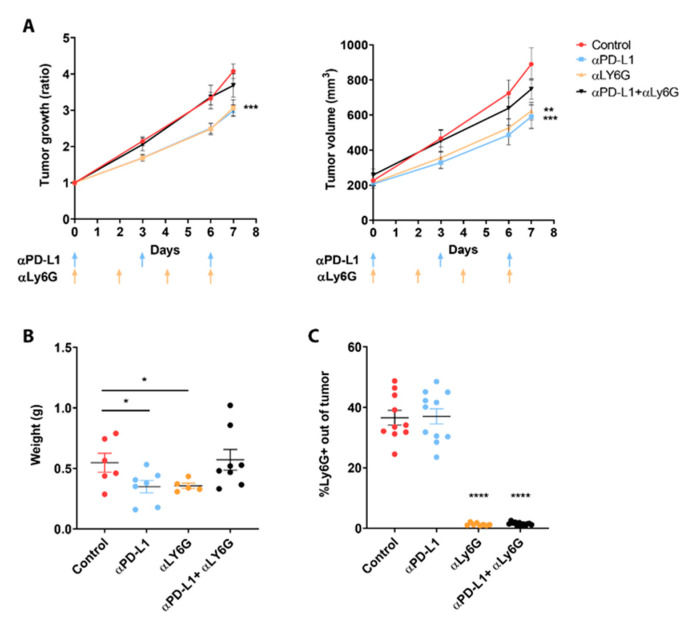
Combination of PD-L1 blockade and neutrophil depletion on tumor development. LLC-GCSF tumor-bearing mice were treated with 250 µg α-PD-L1 and/or 150 µg α-Ly6G antibodies. (**A**) Tumor growth ratio was calculated based on tumor size on each day vs. size day 0 and was assessed following different treatments (left panel). Tumor volume is presented following the treatments (right panel). Days of injections are marked by arrows. (**B**) At day 7, tumors were harvested and weighed. (**C**) The proportion of intertumoral neutrophils were confirmed by quantifying Ly6G+ cells out of whole tumors using FACS. Means values ± SE are shown, *n =* 8–12 each group per experiment. Statistical significance was determined by one-way ANOVA with sidak’s post hoc test (**A**) or unpaired two-tailed *t*-test (**B**,**C**), both with *p* < 0.05. Statistical differences between each treatment and untreated control group are expressed with stars (*); * *p* < 0.05, ** *p* < 0.01, *** *p* < 0.001 and **** *p* < 0.0001.

**Figure 3 cells-11-00783-f003:**
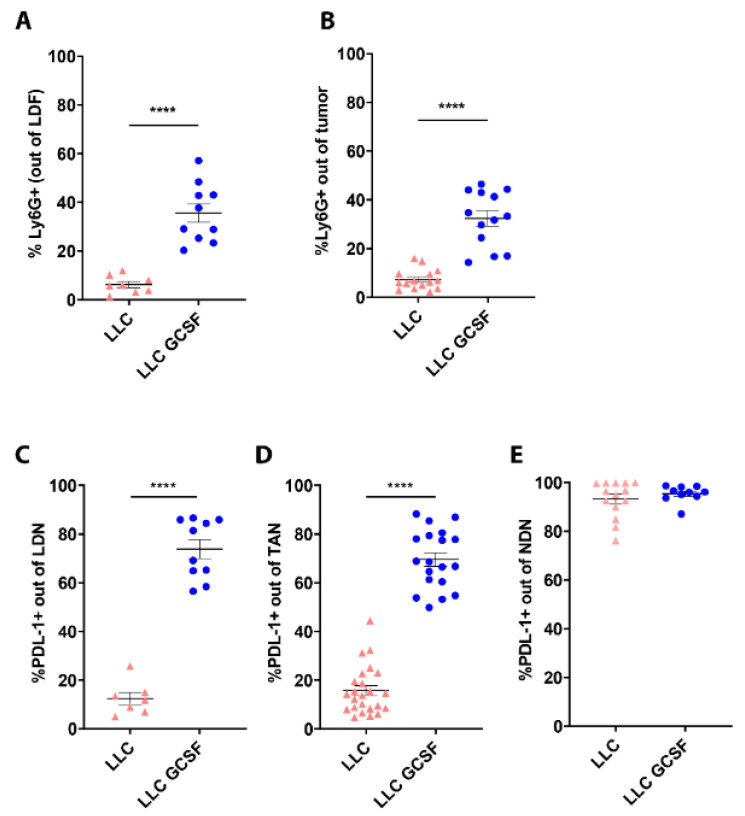
Proportion and PD-L1 expression levels of circulating and intertumoral neutrophil subpopulations in LLC vs. LLC-GCSF models. NDN and LDF were isolated from the circulation of LLC and LLC-GCSF tumor-bearing mice and the tumors were harvested and digested. (**A**,**B**) The proportion of neutrophils in LDF (**A**) and whole tumor (**B**) was quantified based on Ly6G+ cells by flow cytometry. (**C**,**E**) The expression of PD-L1 was assessed in LDN (**C**), TAN (**D**) and NDN (**E**) by flow cytometry. Means values ± SE are presented, *n =* 7–30. Statistical significance was determined by unpaired two-tailed *t*-test with *p* < 0.05. **** *p* < 0.0001.

**Figure 4 cells-11-00783-f004:**
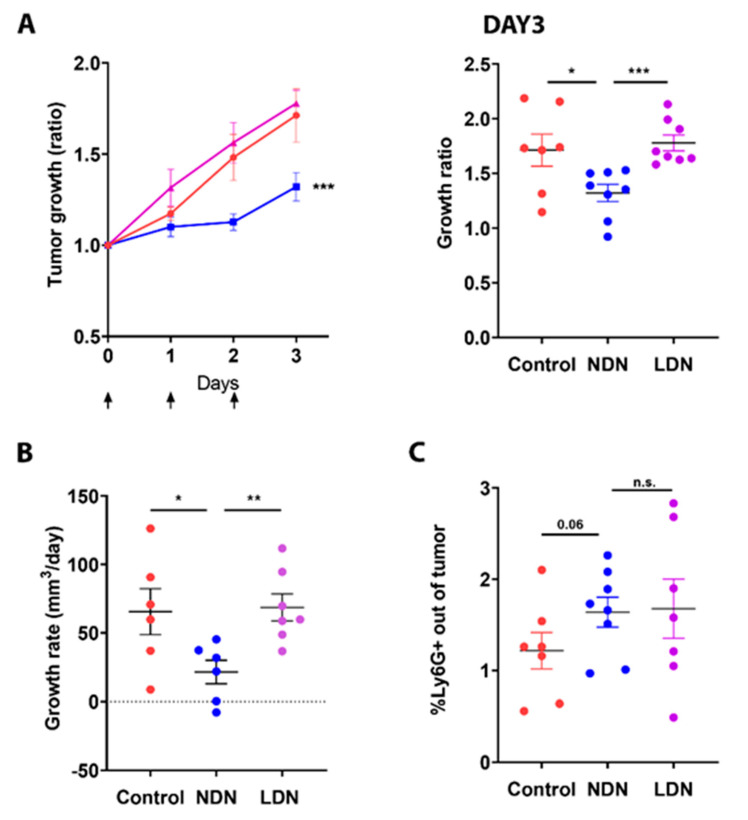
The impact of modulating the NDN/LDN ratio on tumor growth. Mice bearing Lung K-Ras modified (LKRM) tumors were injected with 2–5 × 10^6^ NDN or 5 × 10^6^ LDN. Tumor growth was monitored along the treatments. Days of injections are marked by arrows. (**A**) Tumor growth ratio was calculated based on tumor size on each day vs. size day 0 and was assessed following different treatments over 3 days (Left panel). Tumor growth ratio at day 3 for each mouse is presented (Right panel). (**B**) Individual growth rate (mm^3^/day) was calculated for each mouse. (**C**) The proportion of TAN out of whole tumor following NDN or LDN adoptive transfer was assessed by flow cytometry. Mean values ± SE are shown, *n =* 6–8 in each group per experiment. Statistical significance was determined by one-way ANOVA with sidak’s post hoc test ((**A**)—Left panel), and by unpaired one/two-tailed *t*-test ((**A**)—Right panel, (**B**,**C**)), both with *p* < 0.05. * *p* < 0.05, ** *p* < 0.01, *** *p* < 0.001 and n.s—non-significant.

**Figure 5 cells-11-00783-f005:**
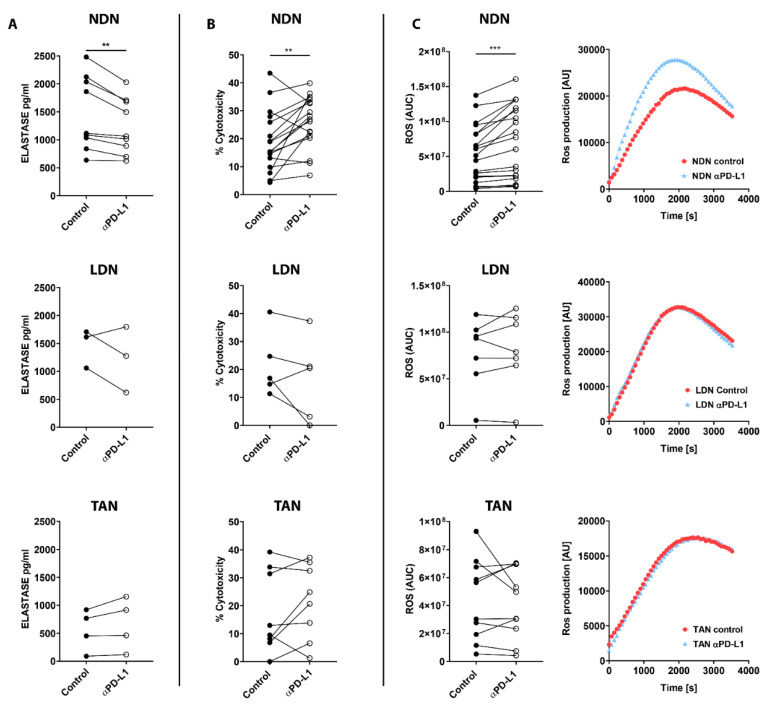
The impact of α-PD-L1 treatment on neutrophils’ functions. NDN, LDN and TAN were isolated from LLC-GCSF tumor-bearing mice and cultured in the absence or presence of 10 µg/mL α-PD-L1 for 4 h (**A**,**C**) or overnight (**B**). (**A**) Neutrophil degranulation was reflected by elastase secretion. Elastase levels were checked by ELISA. (**B**) Neutrophil cytotoxicity was assessed following co-culture with LLC Luciferase tumor cells in ratio of 10:1 overnight. (**C**) ROS production assay was determined by Luminol, HRP assay, following neutrophil activation with 50 nM PMA. Chemiluminescence was measured over a time course of 1 h. Measured chemiluminescence corresponds to ROS production. AUC is presented in left panels. Representative graphs of ROS production over time for each neutrophil subpopulation is provided (Right panels). Means ± SE values are shown. Statistical significance was determined by paired two-tailed *t*-test with *p* < 0.05 Statistical. Differences between each treatment and control are expressed with a star (*). (**A**) *n =* 3–9, ** *p* < 0.01. (**B**) *n =* 5–17, ** *p* < 0.01. (**C**) *n =* 7–20, *** *p* < 0.001.

**Figure 6 cells-11-00783-f006:**
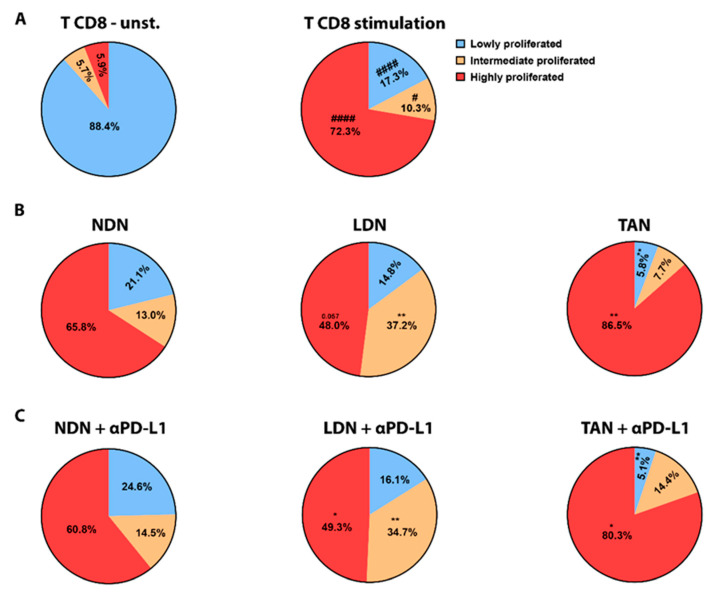
The impact of α-PD-L1 treatment on neutrophil subpopulations’ ability to modulate CD8+ T cells proliferation. CD8+ T cell proliferation was quantified using CFSE labeling assay. The degree of proliferation in T cells was defined by separating whole CD8+ T cell population into 3 groups based on CFSE levels, i.e., low, intermediate and highly proliferated. (**A**) Stimulation with α-CD3 and α-CD28 induces proliferation (right panel) compared to (left panel) unstimulated cells (unst.) Statistical significance was determined by paired two-tailed *t*-test with *p* < 0.05 and is expressed with pound sign (^#^). ^#^
*p* < 0.05 and ^####^
*p* < 0.0001. (**B**) CD8+ T cell proliferation was assessed following CD3/CD28 stimulation and co-culture with NDN, LDN, or TAN for 72 h (**C**) CD8+ T cell proliferation was assessed following CD3/CD28 stimulation and co-culture with NDN, LDN, or TAN for 72 h in the presence of α-PD-L1 antibody. Mean ± SE values are shown, n = 3–26. Statistical significance (**B**,**C**) was determined by one-way ANOVA with Dunnett post hoc corrections with *p* < 0.05. Differences between each co-culture settings, i.e., in the presence of each neutrophil subpopulation (**B**) and in the presence of α-PD-L1 (**C**), vs. CD8+ T cells alone activated with α-CD3 + α-CD8 control are expressed with stars (*). * *p* < 0.05, ** *p* < 0.01.

## Data Availability

Not applicable.

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
