# Peer review of "The Bilateral Interplay between Cancer Immunotherapies and Neutrophils’ Phenotypes and Sub-Populations"

_cells, 2022, doi:10.3390/cells11050783_

Round 1

Reviewer 1 Report

This is an interesting article about the role of neutrophils in cancer.  

I would like to address a small number of suggestions to you.

General recommendations

Please correct all references according to journal stile.

References should be described as follows:

Journal Articles:

  1. Author 1; Author 2. Title of the article. Abbreviated Journal Name Year; Volume: page range.

Introductions

Page 2

Please write abbreviation for Normal-density neutrophils (NDN), low-density neutrophils (LDN), and tumor-associated neutrophils (TANs) in the same manner. If you used plural (Ns) for TANs, it is better to used also for NDN and LDN in this manuscript.

Results

Page 5

The paragraph: "G-CSF has been described as an important factor driving the release of neutrophils from the bone marrow [31], but also capable of affecting myeloid cells’ phenotype [34, 35] and modulating the tumor microenvironment [36, 37].We therefore aimed to assess the impact of G-CSF on tumor growth in the LLC model, and appraise whether the presence of G-CSF could modify the efficacy of immune checkpoint blockade in this model. We used the original LLC cell line and an LLC-GCSF-transfected cell line (referred to as LLC-211 GCSF) secreting high amounts of G-CSF " should be transferred from results section to materials and methods.

Page 6

Line 240 please write in italic “in vivo”

Conclusions are too short.

Author Response

This is an interesting article about the role of neutrophils in cancer.  

Thank you for regarding our manuscript as interesting.

I would like to address a small number of suggestions to you.

General recommendations

Please correct all references according to journal stile.

References should be described as follows:

Journal Articles:

  1. Author 1; Author 2. Title of the article. Abbreviated Journal Name Year; Volume: page range.

References were corrected according to journal style as requested.

Introductions

Page 2

Please write abbreviation for Normal-density neutrophils (NDN), low-density neutrophils (LDN), and tumor-associated neutrophils (TANs) in the same manner. If you used plural (Ns) for TANs, it is better to used also for NDN and LDN in this manuscript.

Thank you for this comment. For maintaining uniformity, we changed TANs to TAN throughout the manuscript, and in the figures.

Results

Page 5

The paragraph: "G-CSF has been described as an important factor driving the release of neutrophils from the bone marrow [31], but also capable of affecting myeloid cells’ phenotype [34, 35] and modulating the tumor microenvironment [36, 37].We therefore aimed to assess the impact of G-CSF on tumor growth in the LLC model, and appraise whether the presence of G-CSF could modify the efficacy of immune checkpoint blockade in this model. We used the original LLC cell line and an LLC-GCSF-transfected cell line (referred to as LLC-211 GCSF) secreting high amounts of G-CSF " should be transferred from results section to materials and methods.

Thank you for this comment. As requested, we moved this paragraph from results section to materials and methods (page 3).

Page 6

Line 240 please write in italic “in vivo”

We confirmed all "in vivo" and "in vitro" throughout the manuscript to be in italic.

Conclusions are too short.

As suggested, we now expanded the conclusions' part at the end of the manuscript (page 15).

Reviewer 2 Report

In their manuscript Kaisar-Iluz investigate comprehensively the role of neutrophils during cancer therapy by check point inhibitors. Overall this is a very important issue. The manuscript is therefore well within the scope of the journal.

Two very minor points in M&M should be adressed:

HRP should be spellt out the first time it is used.

The verctor for G-CSF should be described better, since the cells are sorted for GFP but the source GFP is unclear.

Author Response

In their manuscript Kaisar-Iluz investigate comprehensively the role of neutrophils during cancer therapy by check point inhibitors. Overall this is a very important issue. The manuscript is therefore well within the scope of the journal.

Thank you for your regarding our manuscript as comprehensive and important.

Two very minor points in M&M should be adressed:

HRP should be spellt out the first time it is used.

Thank you for this comment. As requested, HRP was spelled out at the first-time use (mid page 4).

The verctor for G-CSF should be described better, since the cells are sorted for GFP but the source GFP is unclear.

Thank you for this comment. As requested, we now describe in more details the vector used for G-CSF (page 3).